# Body-Mass-Index Associated Differences in Ortho- and Retronasal Olfactory Function and the Individual Significance of Olfaction in Health and Disease

**DOI:** 10.3390/jcm9020366

**Published:** 2020-01-29

**Authors:** Gerold Besser, Brigitte Erlacher, Kadriye Aydinkoc-Tuzcu, David T. Liu, Eleonore Pablik, Verena Niebauer, Martin Koenighofer, Bertold Renner, Christian A. Mueller

**Affiliations:** 1Department of Otorhinolaryngology and Head and Neck Surgery, Medical University of Vienna, Währinger Gürtel 18-20, 1090 Vienna, Austria; gerold.besser@meduniwien.ac.at (G.B.); david.liu@meduniwien.ac.at (D.T.L.); verena.niebauer@gmx.at (V.N.); martin.koenighofer@meduniwien.ac.at (M.K.); 2Medical Department, Krankenhaus der Barmherzigen Schwestern, Stumpergasse 13, 1060 Vienna, Austria; brigitte.erlacher@vinzenzgruppe.at; 3Medical Department, Wilhelminenspital, Montleartstrasse 37, 1160 Vienna, Austria; kadriye.aydinkoc-tuzcu@wienkav.at; 4Section for Medical Statistics, CeMSIIS, Medical University of Vienna, Spitalgasse 23, 1090 Vienna, Austria; eleonore.pablik@meduniwien.ac.at; 5Institute of Experimental and Clinical Pharmacology and Toxicology, University of Erlangen-Nürnberg, 91054 Erlangen, Germany; bertold.renner@tu-dresden.de; 6Institute of Clinical Pharmacology, Medical Faculty Carl Gustav Carus, Technical University Dresden, 01307 Dresden, Germany

**Keywords:** anosmia, flavor, hyposmia, obesity

## Abstract

Odor (including flavor) perception plays a major role in dietary behavior. Orthonasal olfactory function (OOF) has been shown to decrease in obese subjects. Changes in retronasal olfactory function (ROF) after weight loss and in the individual significance of olfaction (ISO) in obesity are yet to be investigated. Firstly, 15 obese subjects were recruited in a pilot study and supported to conventionally lose weight. OOF (Sniffin’ Sticks) was measured at the beginning and after 5.6 ± 1.3 months. Eleven subjects re-visited but barely lost weight and no major changes in OOF were observed. Secondly, the body-mass-index (BMI), OOF, and ROF (Candy Smell Test, CST) were recorded in subjectively olfactory-healthy subjects (SOHSs) and additionally the ISO questionnaire was collected in patients with olfactory dysfunction (OD). BMI correlated significantly negatively with odor discrimination (*p* = 0.00004) in 74 SOHSs and negatively with CST (*p* < 0.0001) in 66 SOHSs. In 48 SOHSs, there was a gender difference in ISO scores (*p* = 0.034), but no significant correlation with BMI was found (*p* > 0.05). ISO scores were significantly higher in 52 OD patients in comparison to SOHSs (*p* = 0.0382). Not only OOF but also ROF may decline with higher BMI. ISO does not seem to alter with BMI, but olfaction becomes more important once it is consciously impaired.

## 1. Introduction

Obesity is a rapidly growing pandemic with the important feature of being preventable [1]. Its causes are multifactorial and lie beyond the obvious reasons of overconsumption of calorie-dense foods and beverages or reduced physical activity. For instance, imbalanced central processing (similar to drug addiction) in regions associated with reward, motivation and conditioning are believed to be causative in obesity development [2]. Additionally, leptin-resistance (an adipose tissue hormone which regulates energy balance) may predispose or promote obesity [3]. Moreover, the leptin hormone may negatively affect olfaction in humans [4] and has been shown to reduce odor discrimination abilities in rodents [5].

As humans rely on their chemosensory abilities to evaluate palatability, smell and taste functions play a major role in appetite, food choice and intake [6]. Olfactory neurons are believed to co-modulate the peripheral metabolism as shown in animal models [7]. Presumable in consequence, olfactory dysfunctions can result in changes in dietary behavior [8]. At the same time, bodyweight negatively correlates with orthonasal olfactory abilities, while studies on retronasal odor perception (i.e., flavor) are still lacking [9]. Whether reduced olfactory abilities are co-causative for obesity or obesity causes an olfactory decline in humans is debatable. Previous investigations showed olfactory abilities to decrease in mice models with weight gain based on fatty nutrition, but this was not reversible when losing weight afterwards [10]. In contrast, bariatric surgeries in humans (in particular sleeve gastrectomy) seem to increase olfactory function [9]. Yet, to our knowledge, studies are missing which investigate whether a non-surgical reduction in bodyweight in humans affects olfactory capacities. 

The sense of smell plays a diverse role in the everyday life of individuals. Odor experts like sommeliers and perfumers evidently pay more attention to this sense [11], and gender differences in the subjective significance of olfaction have been reported [12,13]. In this context, the question arises as to whether individuals with a higher body mass index (BMI) show differences in the awareness of the sense of smell compared to normal-weight subjects.

Thus, the current investigation aimed to address these gaps in knowledge (in particular, retronasal olfaction in obesity and the subjective importance of olfaction) in a prospective trial. The aims of this trial were to (i) assess the comparability of data to previous findings on BMI and olfactory capacity, (ii) to detect weight-loss-associated changes in olfactory abilities in this non-surgical program and (iii) to evaluate the individual significance of olfaction in regard to BMI.

## 2. Materials and Methods

The study was performed according to the Declaration of Helsinki regarding biomedical research involving human subjects, and the study protocol was approved by the local ethics committee (EK-No.: 1335/2017). Written informed consent was obtained.

### 2.1. Pilot Study and Data Collection

In a pilot study, subjects with a BMI equal to or above 30 were recruited between July 2017 and February 2019 through two dietary medical outpatient clinics. At these clinics, patients are supported medically to lose weight in a conventional (non-surgical) approach. For instance, patients are encouraged to leave out in-between dishes (three meals a day), cut down on carbohydrates (especially none in the evening), have salads and vegetables for lunchtime and increase physical activity. Additionally, the medical staff manages co-morbidities such as high blood pressure and diabetes. Measurable changes in bodyweight depend on patients’ compliance.

Sample size calculations were not feasible due to a lack of previous data in this field, however, a study in bariatric surgery showed significant changes in olfactory tests after 3 months in 54 subjects, accompanied by an excess weight loss of 44.9% ± 12.4% [14]. Although expecting less weight loss in a 5-month period of a conventional diet, we planned to include a similar number.

Recruitment stayed far behind expectations, and eventually 15 patients (11 females, 4 males; mean age 43.7 ± 11.2, range 23–63 years) participated. Subjects, in general, stated to have occupations with low physical activity and/or exposure to foods. Four subjects were diabetics. Olfactory tests (see below) were performed initially and after 5.6 ± 1.3 months.

The questions raised above (in particular, aims (i) and (iii)) could not be addressed sufficiently with the obtained data in the pilot study due to the low recruitment. Our group routinely recorded the BMI and provided patients with the questionnaire on the individual significance of the sense of smell (see below). Olfactory test results were part of a recent publication [15]. For retrieval of this data, permission was granted by the local ethics committee (amendment 08/2019). Subjectively olfactory-healthy subjects were recruited via invitational notices displayed at the university campus. Patients with olfactory dysfunction (OD) were recruited through our smell and taste clinic. These patients sought help due to olfaction-related complaints and were tested below the TDI cut-off for normosmia (the TDI score is the summed score of odor threshold (T), discrimination (D) and identification (I) scores). The BMI was used to categorize subjects into “normal weight” (>18.5 and <25.0), “overweight” (>25) and “obese” (>30) [16].

### 2.2. Olfactory Tests

Orthonasal olfactory abilities were assessed using Sniffin’ Sticks (commercially available felt-tip pens, Burghart GmbH, Wedel, Germany). A summed score of three olfactory “dimensions” (odor threshold, T; odor discrimination, D; odor identification, I) was compared with large-population normative data sets [17]. Administration of the three subtests is described in detail elsewhere [18,19,20]. The TDI score can be used to categorize anosmia (16 or less), hyposmia (more than 16, less than 30.75) and normosmia (equal or above 30.75) [17]. Participants were tested in a well-ventilated room and had to refrain from drinking (except water), eating and smoking for at least 30 min prior to testing.

Retronasal olfactory function was tested using the Candy Smell Test (CST) [21,22]. Originally validated in a 23-item version [21], we applied a 27-item version in all subjects [15,23]. For this version, less-identifiable aromas of the original items [21] were removed (three aromas: passion fruit, strawberry and kiwi) and others introduced (seven aromas, see Table 1.) The candies (approximate diameter: 9 mm) contained 500 mg sorbitol and one target aroma. Aromas (obtained from Frey&Lau GmbH, Henstedt-Ulzburg, Germany) were of food-grade quality and were food-related (predominantly sweet-fruity character). Candies (with sorbitol as the candy matrix) are applicable in diabetes, but usage in fructosemia is not recommended [21]. For testing, after placing the candy on the tongue, subjects were asked to suck or chew the candy and name the target aroma out of a list of four possible answers (without visual cues) in a forced-choice manner. After each candy subjects had to rinse their mouth with water. The maximal attainable test score was 27 for the applied 27-item version.

### 2.3. Individual Significance of Olfaction

The original questionnaire on the individual significance of olfaction (ISO) consists of 20 questions [12] and has been applied in studies beyond the introductory publication [13,24,25,26]. Answers are graded from 0 (I totally disagree) to 3 (I totally agree) and a summed score can be obtained with a maximum of 60 points. For the purposes of this study, we extended the questionnaire by 18 novel questions, including 7 more questions with a food and cooking theme (see Table 2). Hence for all applied questions, a summed total score of 114 was possible, with higher scores indicating a higher level of significance (i.e., importance) of the sense of smell to individuals.

### 2.4. Statistical Analysis

IBM SPSS 24.0 (IBM Corp., Armonk, NY, USA) and GraphPad Prism 8.2.0 (GraphPad Software, Inc., La Jolla, San Diego, CA, USA) were used for statistical analysis. Graphical visualization was performed using the same GraphPad Prism. R version 3.5.1 (R Foundation for Statistical Computing, Vienna, Austria) was used for adjustment analysis for age and gender, using a multivariable regression model. Normality of data was tested using the Shapiro–Wilk test. Depending on the normality of data, group differences were tested using (paired or unpaired) sample *t*-test or Mann–Whitney test (for between-subject variables). Equality of variances was explored with Levene’s test and Welch’s correction was applied if necessary. Parametric data are presented as the mean and standard deviation of the mean (SD), as indicated. Correlational analyses were performed using the Pearson correlation coefficient (r). A *p*-value < 0.05 was required for statistical significance. BMI was calculated by dividing measured body weight (kg) by measured height (m) squared.

## 3. Results

### 3.1. Pilot Study 

Table 3 shows olfactory test results at enrollment. Results did not correlate significantly with BMI nor with age (*p* > 0.05). Eleven subjects re-visited, of which only four subjects lost weight since the first visit (mean reduction in BMI: 0.84 ± 2.71, median: −0.27). Orthonasal olfactory performance improved in two cases in a significant way, as defined by previous authors for the TDI (>5.5) [27]. Changes in TDI scores did not correlate significantly with changes in BMI (*p* > 0.05). 

### 3.2. BMI and Olfaction 

Odor identification scores and BMI were collected in 185 subjectively olfactory-healthy subjects (122 females, 63 males; mean age 31.1 ± 12.4, range 19–79 years). Identification scores had a weak, significantly negative correlation with BMI (*r*_185_ = −0.16, *p* = 0.026).

Full TDI scores were collected in 74 subjectively olfactory-healthy subjects (47 females, 27 males; mean age 32.8 ± 11.9, range 20–63 years). BMI correlated significantly negatively with odor discrimination (*r*_74_ = −0.46, *p* = 0.00004) but not significantly with other subtests or TDI (*r*_74_ = −0.140, *p* = 0.234). Table 4 shows the subjects’ characteristics and olfactory results.

Data on CST scores and BMI were available in 66 subjectively olfactory-healthy subjects (42 females, 24 males; mean age 30.9 ± 10.8, range 20–59 years). CST correlated significantly negatively with BMI (*r*_66_ = −0.52, *p* < 0.0001). CST scores in “normal weight” subjects (*n* = 42) differed significantly from “obese” subjects (*n* = 10) (*p* = 0.0035), while “overweight” CST scores (*n* = 14) differed significantly from “obese” scores (*p* = 0.0149) (see Figure 1). When grouping subjects into BMI ≥ 28 (*n* = 11) and <28 (*n* = 55) (as previous investigators did in the field of chemosensory perception and body weight [28]), significantly different CST scores were found (*p* = 0.019).

Results were adjusted for age and gender: age had no significant influence on CST nor on odor discrimination (D). For the variable “gender,” a trend towards higher CST and D scores was seen in males, but this did not reach statistical significance at alpha 0.05.

### 3.3. Individual Significance of Olfaction 

Data on the ISO and BMI were available in 48 subjectively olfactory-healthy subjects (30 females, 18 males; mean age 33.3 ± 12.4, range 22–63 years) and in 52 patients with OD (35 females, 17 males; mean age 55.9 ± 17.7, range 21–83 years). In these OD patients, 23 (44.2%) were of normal weight, 24 (46.2%) were overweight and 5 (9.6%) were obese, partly aligning with previous findings proposing a high likelihood of patients with OD being overweight and obese [29].

In subjectively olfactory-healthy subjects, ISO scores differed significantly in male and female subjects (*p* = 0.034), indicating a different subjective value of the sense of smell in male and female subjects and aligning with previous findings in different cultures [13]. ISO scores did not significantly correlate with BMI (*r*_48_ = 0.204, *p* = 0.164), nor was there a BMI group difference. Table 5 shows the subjects’ characteristics and questionnaire results.

Overall, subject scores on the original 20 questions and scores on the new 18 questions (see Table 2) correlated strongly (*r*_100_ = 0.754, *p* < 0.001). Summed ISO scores were significantly higher in patients with olfactory dysfunction in comparison to subjectively healthy subjects (*p* = 0.0382), suggesting a higher subjective significance of the sense of smell when it is impaired (see Figure 2). The extended questionnaire (38 questions) offered an excellent internal consistency (Cronbach’s Alpha = 0.921).

## 4. Discussion

As a major finding of the present work, in addition to orthonasal odor perception, retronasal flavor identification abilities also declined with rising BMI. Scores on the applied 27-item Candy Smell Test were significantly lower in subjects with higher BMI. We also found significant BMI group differences on the CST. Furthermore, an unnoticed decline in olfactory performance in subjects with higher BMI did not go together with a subjective altered “vision” of the sense of smell (i.e., olfaction was not more or less important in obese subjects). In contrast, once olfactory dysfunction was noticed, awareness of the sense of smell was more pronounced, as mirrored by higher scores on the ISO.

This investigation addressed several questions in chemosensory perception and obesity. Research progress in this field derives mostly from experimental studies in rodents [7,10,30] and there is a clear need for clinical investigations in humans—especially since olfactory testing methods (e.g., retronasal tests) have become more standardized and hence comparable [17,21,31]. Notably, the number of publications on bodyweight and multiple olfactory dimensions is still low and information on the relation of retronasal olfactory tests and BMI is still missing [9]. 

In a previous study on olfaction and obesity, the authors found weak correlations between TDI subtest scores and BMI [32]. Our results on identification scores aligned with this, but discrimination scores and retronasal test results showed clearer negative associations with rising BMI (also when adjusted for age and gender). To our best knowledge, this finding of decreasing retronasal abilities with increasing BMI is a novel aspect, which may have profound clinical implications in obesity and the treatment of obese patients.

As mentioned previously, the candies applied in this study consisted of a sorbitol matrix and therefore had a sweet character. However, individual sweet sensitivity did not interfere with CST results: all participating subjects did not report taste impairment (therefore taste dysfunction is unlikely [33]) and sweet thresholds in particular seem to be unchanged in obese adults [34]. Nevertheless, we encourage future investigators to consider also including flavors of non-sweet-fruity characters (as proposed and cross-culturally applied by previous authors [31,35]) to further elucidate the weaknesses of flavor perceptive abilities in obese subjects.

“Flavor” perception, and thus food enjoyment, strongly relies on retronasal olfaction [36], whereas “taste” perception refers to five basic qualities (salty, sweet, sour, bitter and umami) detected by receptors on the tongue and soft palate [37]. Diminished perceptive abilities through the retronasal route make food taste dull, as commonly experienced during a common cold with a stuffed nose. Due to this, it could be postulated that obese individuals are less capable of enjoying food (i.e., flavors) and need more of it to satisfy their needs. Previous authors have also shown associations of poorer odor-identification and fat-discrimination abilities with a diet rich in saturated fats and added sugar (i.e., an unhealthy diet) [38]. Obesity management remains challenging and varies throughout countries, despite available guidelines [39]. All the more novel treatment options are valuable. For instance, intermittent fasting seems to be able to provide a metabolic benefit which, among other things, decreases leptin levels [40]. In regard to possibly reduced flavor-perception abilities in obesity, a healthy and balanced cuisine that also activates other sensory systems (e.g., trigeminal activation through spicy food or food that is variable in texture) may be useful in this context to increase enjoyment and at the same time decrease the amount of food needed. Furthermore, since odors can be both appetizing (for the odor-cued food) and satiating (for the non-odor-cued food) [41], the question raises if odor stimulation can modulate the reward system. 

Food choice is strongly affected by chemosensory perceptive abilities. Broad research efforts have been put into the relations between taste function and bodyweight [9], but evidently, flavor perception may influence food liking (and hence food choice) more markedly. Liking and reward value seem to be dependent on whether the food has been eaten recently or not [42] and if flavors have been encountered before. As a consequence of this study’s findings, it can be hypothesized that decreased flavor-identification abilities with rising BMI may negatively influence food choice. This is emphasized by recent functional MRI findings suggesting a blunted reward response in obesity [43]. Moreover, flavor liking seems to differ among weight groups, as shown for different butter aroma concentrations in normal-weight compared to obese subjects [44]. Therefore, further studies exploring the impact of flavor perception on food choice, the reward system and metabolism are warranted.

One limitation concerning the olfactory findings of the present investigation needs to be highlighted: despite utilizing a broad spectrum of standardized olfactory tests, there was an unbalanced number of subjects among BMI groups in this data set. Additionally, the initial aim to investigate possible changes in olfactory function in the course of weight loss was not feasible because of low recruitment rates and evident low dietary compliance (close to no major changes in the bodyweight of revisiting subjects). In bariatric surgery, not weight loss but effects on the vagal nerve are suggested to be the reason for changes in olfactory performance [9]. In order to underpin this hypothesis, further prospective trials are needed in obese humans willing to lose weight. Recruitment and compliance difficulties in the present trial may have been due to the location of the study sites. Medical counseling took place elsewhere and subjects had to voluntarily travel within the city to our lab for olfactory testing. Multicenter approaches and onsite olfactory testing should be attempted in order to obtain sufficient data.

In summary, not only orthonasal but also retronasal olfactory abilities may decline with higher BMI. Furthermore, the subjective importance of the sense of smell does not seem to alter with BMI, but it becomes more evident once the sense of smell is consciously impaired, as it is in patients with olfactory dysfunction. Further studies on bodyweight and olfaction, using established retronasal tests, are needed.

## Figures and Tables

**Figure 1 jcm-09-00366-f001:**
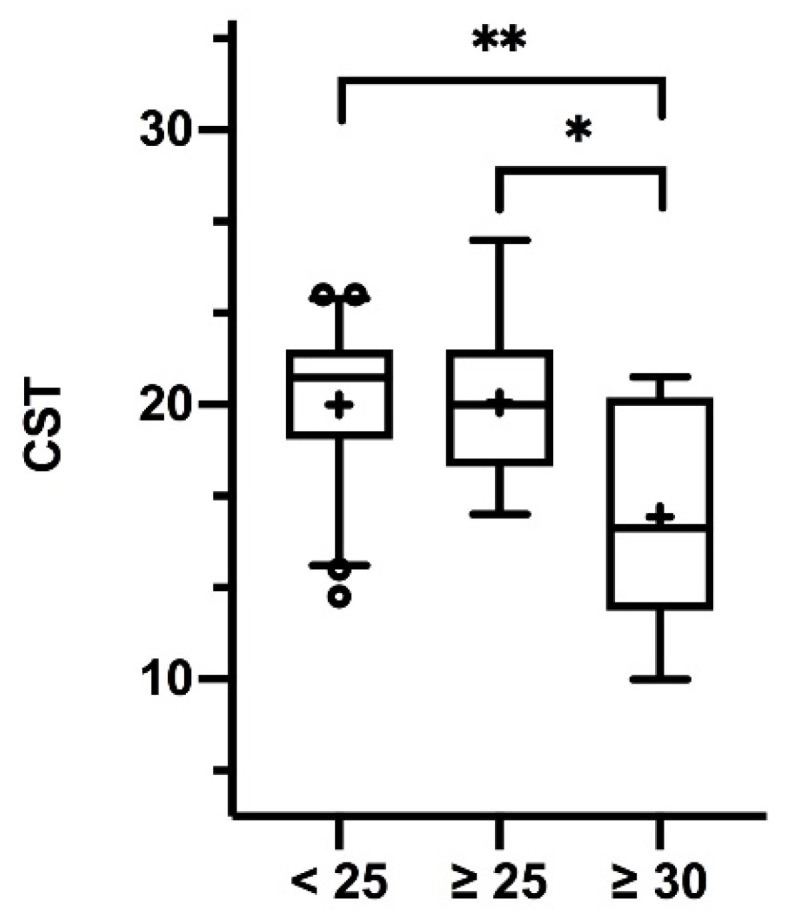
Box-and-whisker plots of retronasal olfactory test results (CST) per body mass index (BMI) group (normal weight < 25, overweight ≥ 25 and obese ≥ 30). Medians (Q0.5; line), interquartile range (Q0.25, Q0.75; boxes); + mean scores; ° outliers. Significant group differences: * *p* = 0.0149, ** *p* = 0.0035.

**Figure 2 jcm-09-00366-f002:**
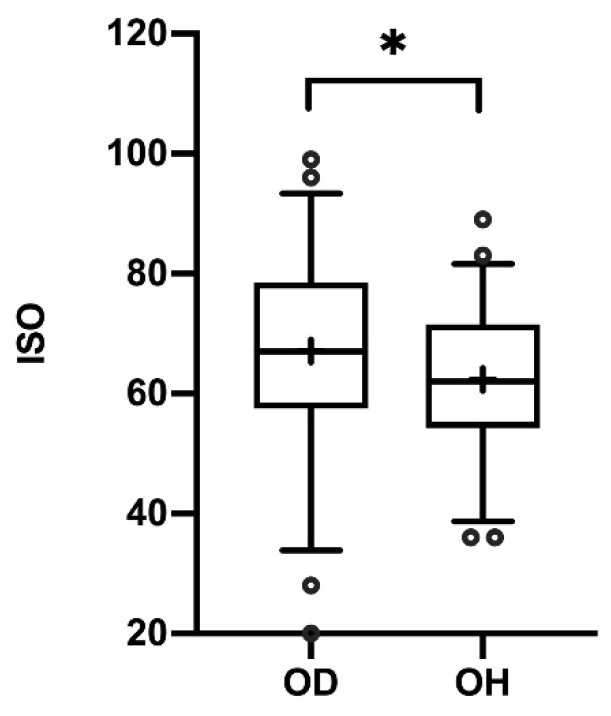
Box-and-whisker plots of scores on the questionnaire of individual significance of olfaction (ISO). Medians (Q0.5; line), interquartile range (Q0.25, Q0.75; boxes); + mean scores; ° outliers. Scores of subjects with olfactory disfunction (OD) and subjectively olfactory-healthy (OH) subjects differed significantly: * *p* = 0.0382.

**Table 1 jcm-09-00366-t001:** The 27-item Candy Smell Test (CST).

No.	Item 1	Item 2	Item 3	Item 4
1	**lemon**	apple	peppermint	gingerbread
2	chocolate	**raspberry**	coconut	cherry
3	walnut	tomato	**coke**	peach
4	**cinnamon**	pear	apple	woodruff
5	cucumber	chocolate	**banana**	orange
6	grapes	vanilla	hazelnut	**orange**
7	lemon	cube sugar	**pineapple**	nut nougat
8	honeydew	**coffee**	strawberry	mandarin
9	orange	coke	**peach**	chocolate
10	**woodruff**	chocolate	peanut	strawberry
11	cucumber	cinnamon	**pear**	licorice
12	banana	**gingerbread**	lemon	grapes
13	coconut	strawberry	walnut	**blackcurrant**
14	mandarin	peanut	**anise**	caramel
15	passion fruit	blackcurrant	**hazelnut**	pineapple
16	**mandarin**	nougat	kiwi	vanilla
17	**apple**	peanut	orange	coke
18	coke	**vanilla**	hazelnut	tomato
19	cucumber	coconut	cube sugar	**cherry**
20	coconut	orange	coke	**apricot**
21	coffee	**rhubarb**	walnut	lemon
22	vanilla	kiwi	**coconut**	orange
23	peppermint	chocolate	**eisbonbon**	tomato
24	**plum**	anise	gingerbread	licorice
25	**condensed milk**	mandarin	walnut	chocolate
26	raspberry	**nut nougat**	honeydew	vanilla
27	licorice	anise	**peppermint**	apple

In bold: target aromas; Not bold: distractors; Novel items: numbers in bold.

**Table 2 jcm-09-00366-t002:** Individual significance of olfaction: novel additional questions.

No.	Statement	A	B	C	D
1	I pay attention to odors in my surroundings when leaving the house				
2	When eating an apple, I think of its smell				
3	When I see flowers, I consciously smell them				
4	On a market, I consciously notice the odors				
5	When cooking, I smell each ingredient to see if they match				
6	When a bedroom smells unpleasant, I let some air in				
7	Thinking of my partner, I think of his/her smell				
8	Smelling a glass of wine, I pay attention to different aromas				
9	I remember body odors of relatives/friends/familiar persons				
10	When buying flowers, I decide by the smell				
11	After a rainfall, I notice odors more intensely				
12	When cooking, I pay attention to the smell of each ingredient				
13	Often during the course of the day, I check if my hands, armpits, breath et cetera smell funny				
14	I notice seasonal (winter/summer) differences in surrounding odors				
15	For my occupation a good sense of smell is essential				
16	For my hobbies a good sense of smell is essential				
17	Good food is my greatest passion				
18	Enjoying good wine makes me happy				

A: I totally agree; B: I tend to agree; C: I rather disagree; D: I totally disagree.

**Table 3 jcm-09-00366-t003:** Subject characteristics and obtained scores at enrollment of the prospective pilot study.

	*n* = 15	
	Mean	SD
Age	43.7	11.2
BMI	44.3	6.2
Odor threshold	6.0	2.5
Odor discrimination	12.1	1.7
Odor identification	13.3	1.8
TDI	31.3	4.7
ISO	65.5	14.1
	*n* = 7	
Candy Smell Test	15.3	4.6

BMI: body mass index; ISO: questionnaire-based assessment of the individual significance of olfaction; TDI: summed score of odor threshold (T), discrimination (D) and identification (I) scores; SD: standard deviation of mean.

**Table 4 jcm-09-00366-t004:** Subject characteristics and obtained olfactory scores.

***n* = 185 (122 f, 63 m)**	**Mean**	**SD**
Age	31.1	12.4
BMI	24.9	7.0
Odor identification	13.8	1.3
***n* = 74 (47 f, 27 m)**	**Mean**	**SD**
Age	32.8	11.9
BMI	27.3	9.5
Odor threshold	6.7	2.8
Odor discrimination	13.6	1.7
Odor identification	13.9	1.3
TDI	33.0	4.5
***n* = 66 (42 f, 24 m)**	**Mean**	**SD**
Age	30.9	10.8
BMI	25.2	7.4
Candy Smell Test	19.4	3.3

BMI: body mass index; TDI: summed score of odor threshold (T), discrimination (D) and identification (I) scores; SD: standard deviation of mean. f: females; m: males.

**Table 5 jcm-09-00366-t005:** Subject characteristics and questionnaire scores.

***n* = 48 (30 f, 18 m) OH**	**Mean**	**SD**
Age	33.3	12.4
BMI	29.8	10.8
ISO	62.3	12.0
***n* = 52 (35 f, 17 m) OD**	**Mean**	**SD**
Age	55.9	17.7
BMI	25.7	4.5
ISO	68.4	15.2

BMI: body mass index; ISO: questionnaire-based assessment of the individual significance of olfaction; OD: subjects with olfactory dysfunction; OH: subjectively olfactory-healthy subjects; SD: standard deviation of mean. f: females; m: males.

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
