# Peer review of "Body-Mass-Index Associated Differences in Ortho- and Retronasal Olfactory Function and the Individual Significance of Olfaction in Health and Disease"

_jcm, 2020, doi:10.3390/jcm9020366_

Round 1
Reviewer 1 Report
Using data from another publication is questionable (lines 95-107). Is it necessary to explain that.
That the "local ethics commission" granted use of published data - is that the task of this committee. And what does a reference to "amendment 08/2019" mean to the reader?
I can only point his out.
Reviewer 2 Report
The authors addressed my main concerns.
This manuscript is a resubmission of an earlier submission. The following is a list of the peer review reports and author responses from that submission.
Round 1
Reviewer 1 Report
jcm-674404
Besser et al.
Body-mass-index…
General
An interesting topic, the manuscript is well-written, but several issues must be addressed prior to publication. I'm insecure whether a major revision could afford the necessary changes..
(1) Does the ms provide data to support the conclusions? I'm insecure, but I might misunderstand things.
The abstract states that "…15 obese subjects were recruited and supported to conventionally lose weight…" (lines 21-22), while their orthonasal olfactory function did not change (lines 24-25).
This is misleading, since these subjects did not loose weight (line 222).
Understandably, it ain't easy to collect data on obese subjects who succeed in reducing weight, but the abstract should clearly state what was done, and which conclusions can be drawn.
Furthermore, the Discussion states in line 191 ff. that "…a major finding of the present work, in addition to orthonasal odor perception also retronasal flavor identification abilities declined with rising BMI". I have a hard time seeing how this ms provides evidence for this statement.
This must be amended, or become much clearer.
(2) Data derived from other/unknown (?) sources.
Admittedly, I might have overlooked or misunderstood where data were obtained that are shown in the Results (section 3.2 and 3.3)
line 151. Data … were retrievable
154. Full TDI scores were retrievable…
157. Data … were available…
172. Data … were available…
Has this data been published? It would be necessary to cite the source. If this is unpublished data, the M&M section should clearly state the source (a database?) and how the data were generated.
This must become clearer.
(3) This study combines several topics, and an effort should be made to very clearly state the aims of the study, and which data was collected/obtained to this purpose. The title announces differences in ortho- and retronasal olfactory function, plus individual differences in olfaction. These are related, yet distinct topics.
Does olfactory training (OT) belong here, and are there enough results to warrant conclusions?
The aims are listed under M&M, section 2.1 - but should really be given in the Intro, instead.
(4) There are also several minor issues. The use of multiple abbreviations doesn't necessarily improve readability. Some terms are simple and short enough to be spelt out, like olfactory training (OT). A complete list of all abbreviations would almost be necessary.
Table 1 mixes experiments and is hard to understand. Are all parameters and units clearly defined and simple enough to make the table readable? There is plenty of space in the table to state full terms plus abbreviations (instead of explaining abbreviations in footer).
Part of the Discussion is not entirely easy to follow, such as lines 211-219.
Author Response
An interesting topic, the manuscript is well-written, but several issues must be addressed prior to publication. I'm insecure whether a major revision could afford the necessary changes.
(1) Does the ms provide data to support the conclusions? I'm insecure, but I might misunderstand things.
The primary conclusion (first and last paragraph discussion) is that also retronasal olfactory function may decline with rising BMI. We found good correlation to support this also with BMI group differences (see Figure 1). Adjustments for age and gender did not contradict this. The questionnaire results revealed more importance for the sense of smell in the group of patients with olfactory dysfunction (see Figure 2). The main difference in these groups is, one is subjectively olfactory healthy, the other one presents to a smell and taste clinic with olfactory related complaints – hence consciously feel their sense of smell to be disturbed. Rising BMI may lead to reduced unnoticed olfactory decline, which does not affect the questionnaire results, while patients with dysfunction score higher on this questionnaire. We added information on this in the Figure 2.-legend (“subjectively”) and extended the last paragraph (“…becomes more evident once the sense of smell is consciously impaired, as it is in patients with olfactory dysfunction.”). We hope this clarifies the situation and improves readability.
The abstract states that "…15 obese subjects were recruited and supported to conventionally lose weight…" (lines 21-22), while their orthonasal olfactory function did not change (lines 24-25). This is misleading, since these subjects did not lose weight (line 222). Understandably, it ain't easy to collect data on obese subjects who succeed in reducing weight, but the abstract should clearly state what was done, and which conclusions can be drawn.
We payed attention to the abstract, as indeed the presentation of the study design may be misleading. We split the M&M section within the abstract by adding “firstly” and “secondly” and moved the findings of the pilot study prior to “secondly”. Information on data pooling is now in the abstract: “Secondly, we gathered yet unpublished data from studies by our group”. This way it becomes clearer what was done (see line 21-27, CleanR1). Efforts were employed to shorten the abstract (e.g., double phrasing “correlated significantly negatively with BMI” was removed and the sentence adjusted appropriately).
Furthermore, the Discussion states in line 191 ff. that "…a major finding of the present work, in addition to orthonasal odor perception also retronasal flavor identification abilities declined with rising BMI". I have a hard time seeing how this ms provides evidence for this statement. This must be amended or become much clearer.
As mentioned, we show good correlation CST/BMI and group differences, which are visualized in Figure 1. To highlight this, we extended the paragraph former line 191 by: “Scores on the applied 27-item Candy-smell-test were significantly lower in subjects with higher BMI. Also, we found significant BMI-group differences on the CST.”
(2) Data derived from other/unknown (?) sources. Admittedly, I might have overlooked or misunderstood where data were obtained that are shown in the Results (section 3.2 and 3.3)line 151. Data … were retrievable154. Full TDI scores were retrievable…157. Data … were available…172. Data … were available… Has this data been published? It would be necessary to cite the source. If this is unpublished data, the M&M section should clearly state the source (a database?) and how the data were generated. This must become clearer.
We thank for the valuable suggestions. Due to low recruitment rates in the course of the study, it became clear, raised scientific questions were not going to be evaluable with obtained data. We therefore proposed amendment to the ethic committee on the original study protocol. The amendment was approved 06thof August 2019. Pooled data is part of ongoing studies or studies of our group with completed data collection. To date, these studies have not been published yet. Investigators shared data in an anonymized fashion. To clarify this, we positioned the “data collection” section earlier in the text. The first paragraph in M&M section now already informs about data pooling: “in a prospective trial. Additionally, pooled data was analyzed (see below).” (line 72, CleanR1). Then 2.2 now includes “from ongoing studies and yet unpublished studies with completed data collection” (line 98, CleanR1) and “For this permission was granted by the local ethics committee (amendment 08/2019).” (line 101, CleanR1). This information is now also present in the abstract (line 24, CleanR1). We hope this now gives the reader sufficient information on where the data came from.
(3) This study combines several topics, and an effort should be made to very clearly state the aims of the study, and which data was collected/obtained to this purpose. The title announces differences in ortho- and retronasal olfactory function, plus individual differences in olfaction. These are related, yet distinct topics. Does olfactory training (OT) belong here, and are there enough results to warrant conclusions? The aims are listed under M&M, section 2.1 - but should really be given in the Intro, instead.
In fact, this study raises several questions and as requested, aims were moved to the Introduction section (see line 63, CleanR1), which introduces raised questions appropriately earlier. In planning phase of the study, olfactory training in a dietary context was of great interest to our group. Data on this is still lacking. This study does not provide enough results to be able to draw sufficient conclusions. Nevertheless, we think the idea is worth a paragraph within this paper to encourage future investigations. We added “and to this point no sufficient conclusions can be drawn from this data” to the discussion segment to underline the limitations of collected data (see also reviewer 2). Also, we do not mention olfactory training in the abstract, as this is not the core massage of the paper.
(4) There are also several minor issues. The use of multiple abbreviations doesn't necessarily improve readability. Some terms are simple and short enough to be spelt out, like olfactory training (OT). A complete list of all abbreviations would almost be necessary.
Thank you for this valuable criticism. We omitted the “OT” abbreviation and also “D” – odor discrimination.
Table 1 mixes experiments and is hard to understand. Are all parameters and units clearly defined and simple enough to make the table readable? There is plenty of space in the table to state full terms plus abbreviations (instead of explaining abbreviations in footer).
Again, thank you for this valuable criticism. We use full terms now where appropriate. With the added Table 2 it should be easier to understand for the reader.
Part of the Discussion is not entirely easy to follow, such as lines 211-219.
We substantially reorganized this segment by adding new information (see also comment on “sweet” above and split it paragraphs. This should be more convenient to follow.
Reviewer 2 Report
Major Comments:
This paper examines retronasal olfactory ability in obese patients. Patients/Subjects were recruited in a variety a manners. Subjects were tested in several olfactory tests and results were binned into BMI categories.
My major concern is in the writing style. This makes the paper confusing to read.
First, the authors need to reconsider how they discuss the effects of olfactory training. Only two patients attempting training, yet it is mentioned in the introduction as if there will be substantive data. In fact, it is difficult to asses anything from those data.
The candy-smell test needs to be better explained. I say this for two reasons. This test is not quite a standard test and the test described here is a modified version of the published test. The methods describe the candies as "...containing 500mg sorbitol and the one targeted aroma." Please describe if each candy had one of 27 novel target aromas or if each had the same aromas. Please also describe the distractor aromas.
Also, why was the CST score only available for 7 of the patients in Table 1?
The methods and the introduction discuss and describe patients in the prospective trial (i.e. Table 1). Yet most of the data (Figures 1 and 2) seems to come from other subjects. A separate table for these patients is necessary.
It is unclear which CST was performed on the 'other' patients. Did the task have 27 or 23 candies? Further, he CST is a flavor task as candies contain a sweet taste (sorbitol) and an aroma (unddefined in the paper). By questioning about the identity of the aroma the task is used to test retronasal olfactory identification. Was sweet perception as measured?
Minor Comments:
Table 1. Misspelling: "Enrollment"
Figure 1 and Figure 2 have an unexplained symbol °. Further, in Figure 1 the symbol is only present in the <25 group.
The so-called TDI test is an acronym with three perceptual components (threshold, discrimination, identification). It makes sense to use the initials in Table 1. However, it is confusing to use the abbreviations in the sentences. For example in line 152-153: "I correlated weakly significantly negatively with BMI (stats)". Is an odd sentence.
The document needs to checked for spelling errors. For example, Line 224 'trails' should be changed to 'trials'.
In the conclusions, the authors claim to have studied dietary behavior (Line 197). I don't agree. Nowhere in the paper are data specifically on food choice, meal duration, meal consumption, etc.
The conclusions could be strengthened by including a few missing references. A few authors have compared retronasal/flavor responses to BMI. Stevenson et al., looked at vanilla https://academic.oup.com/chemse/article/41/6/505/1744924
While, Proserpio investigated vanilla and 'butter flavor' https://www.sciencedirect.com/science/article/pii/S0950329316301999#f0025
And Jacobson compared BMI to fMRI results: https://www.ncbi.nlm.nih.gov/pmc/articles/PMC6520683/
Author Response
# Reviewer 2
Major Comments:
This paper examines retronasal olfactory ability in obese patients. Patients/Subjects were recruited in a variety a manners. Subjects were tested in several olfactory tests and results were binned into BMI categories. My major concern is in the writing style. This makes the paper confusing to read.
We thank the reviewer for the critical comments and improvement suggestions. Following both reviewers’ comments, we substantially changed the manuscript and are confident it is now less confusing to read and in improved in quality. Please see all point-by-point responses.
First, the authors need to reconsider how they discuss the effects of olfactory training. Only two patients attempting training, yet it is mentioned in the introduction as if there will be substantive data. In fact, it is difficult to assess anything from those data.
In planning phase of the study, we were quite confident to recruit enough subjects in this prospective trail to evaluate acceptance of olfactory training. However, this failed and indeed it is difficult to assess a lot from this limited data. Nevertheless, we wanted to spread the idea of olfactory training in a dietary setting to a broad public – hence not cut down on this information – and encourage future investigations. We added “Compliance however was very low (see results).” in the M&M section to underline the limited data on this topic. Also, we added “Compliance in performing OT was rather low and to this point no sufficient conclusions can be drawn from this data.” in the discussion (line 258, CleanR1). We hope this is a reasonable compromise.
The candy-smell test needs to be better explained. I say this for two reasons. This test is not quite a standard test and the test described here is a modified version of the published test. The methods describe the candies as "...containing 500mg sorbitol and the one targeted aroma." Please describe if each candy had one of 27 novel target aromas or if each had the same aromas. Please also describe the distractor aromas.
We substantially changed the paragraph on the Candy-Smell-Test, now containing much more information (please see line 120-130, CleanR1). According to your request we added information on distractors, which was best done in an additional table (Table 3.). This table now illustrates the novel 27-item CST in a comprehensive way for the first time within a paper including each distractor. Normative data on this test is on the way by our group and we expect numerous citations of the present article due to this.
Also, why was the CST score only available for 7 of the patients in Table 1?
We added information on the restrictive use of the CST in fructosaemia (line 127, CleanR1). In fact, 2 subjects of the prospective trial claimed to suffer from fructosaemia and were not tested with the CST. The remaining 6 unfortunately were included earlier in the study than the 27-CST was provided by manufacturing unit. We did not exclude these subjects however due to the small sample size.
The methods and the introduction discuss and describe patients in the prospective trial (i.e. Table 1). Yet most of the data (Figures 1 and 2) seems to come from other subjects. A separate table for these patients is necessary.
The M&M now clearer states were the data comes from and we introduced a separate table (Table 2.) as requested.
It is unclear which CST was performed on the 'other' patients. Did the task have 27 or 23 candies?
This is now made clear in line 121 (CleanR1), all participants performed the 27-item version.
Further, the CST is a flavor task as candies contain a sweet taste (sorbitol) and an aroma (undefined in the paper). By questioning about the identity of the aroma the task is used to test retronasal olfactory identification. Was sweet perception as measured?
Aromas are now precisely defined within Table 2., also novel items are highlighted. Indeed, since the candy matrix is sorbitol, a sweet character accompanies aligned aromas. It can be speculated that changes in sweet sensitivity may interfere with results on this test. However, from our experience, aromas of the candies overexpress the sweet character and we therefore did not measure sweet perception/sensitivity. Furthermore, included subjects did not report taste dysfunction, hence it is very unlikely they suffer from taste dysfunction.[1] Although, research on weight and taste is somehow contradictory, especially sweet thresholds seem to be unchanged in obese adults[2], which is why we do not see a potential bias. We implemented these thoughts in the discussion section including the references (line 227-230, CleanR1).
Soter A, Kim J, Jackman A, Tourbier I, Kaul A, Doty RL. Accuracy of self-report in detecting taste dysfunction. 2008;118(4):611–7. Donaldson LF, Bennett L, Baic S, Melichar JK. Taste and weight: is there a link? Am J Clin Nutr. 2009 Sep;90(3):800S-803S. doi: 10.3945/ajcn.2009.27462Q.
Minor Comments:
Table 1. Misspelling: "Enrollment"
Thank you, corrected.
Figure 1 and Figure 2 have an unexplained symbol °. Further, in Figure 1 the symbol is only present in the <25 group.
Indeed, this is not explained: ° refers to outliers, this now is explained in the figure legend.
The so-called TDI test is an acronym with three perceptual components (threshold, discrimination, identification). It makes sense to use the initials in Table 1. However, it is confusing to use the abbreviations in the sentences. For example in line 152-153: "I correlated weakly significantly negatively with BMI (stats)". Is an odd sentence.
The document needs to checked for spelling errors. For example, Line 224 'trails' should be changed to 'trials'.
Indeed, this may lead to misunderstandings, corrected to: “Identification scores correlated weakly significantly …”. Thank you, corrected: “trials”. Table 1 and 2 now use only necessary abbreviations.
In the conclusions, the authors claim to have studied dietary behavior (Line 197). I don't agree. Nowhere in the paper are data specifically on food choice, meal duration, meal consumption, etc.
This is true, we have to admit. We deleted “dietary behavior”. Future investigations need to elucidate this.
The conclusions could be strengthened by including a few missing references. A few authors have compared retronasal/flavor responses to BMI.
Stevenson et al., looked at vanilla https://academic.oup.com/chemse/article/41/6/505/1744924
While, Proserpio investigated vanilla and 'butter flavor' https://www.sciencedirect.com/science/article/pii/S0950329316301999#f0025
And Jacobson compared BMI to fMRI results: https://www.ncbi.nlm.nih.gov/pmc/articles/PMC6520683/
We thank for these valuable suggestions and implemented these in the discussion section with the following phrases: “Also, previous authors showed associations of poorer odor identification and fat discrimination abilities with a diet rich in saturated fats and added sugar (hence unhealthy diet) [40].”, “Recent functional magnetic resonance imaging findings suggest a blunted reward response in obesity [41].” and “Also, flavor liking seems to differ among weight groups, as shown for different butter aroma concentrations in normal weight compared to obese subjects [43].”.
Round 2
Reviewer 1 Report
My main comments on the original ms version were
(1) Does the ms provide data to support the conclusions?
(2) Data derived from other/unknown (?) sources / Sufficient description of how data were collected/obtained.
(3) This study combines several topics.
-----
I'm insecure whether it is necessary to point out that an original research paper should consist of
- a M&M section where data collection is described in a fashion that would enable others to repeat the same experiments
- a Results section describing the data obtained.
- leading to conclusions, typically given in Abstract and Discussion.
Needless to mention that the data presented must be unpublished data, and that data collection is complete/finalized.
-----
Authors' response to #1
"…Secondly, we gathered yet unpublished data from studies by our group. This way it becomes clearer what was done."
M&M (section 2.4) now reads: "We therefore pooled obtained data with data from ongoing studies and yet unpublished studies with completed data collection"
I find it hard to penetrate/understand this. What is meant with "data pooling"? Is the intention to publish data from "ongoing studies" and "yet unpublished studies" partly here and partly elsewhere, in the next paper?
Does Table 2, which is given under M&M, show data? Then it should be moved to the Results section? Also - the Table header reads "Pooled data…". What does this mean? I still can't find a description of how this data was collected.
Authors' response to #2
"Pooled data is part of ongoing studies or studies of our group with completed data collection. To date, these studies have not been published yet. Investigators shared data in an anonymized fashion. To clarify this, we positioned the “data collection” section earlier in the text."
"Then 2.2 now includes “from ongoing studies and yet unpublished studies with completed data collection”
I'm puzzled (and don't agree that this issue has been clarified).
Data from "studies [that] have not been published yet" sounds like the idea is to publish this data later on, elsewhere?
The data should either be shown/published here or deleted from this current ms (and saved for another paper). However, it ain't teh right strategy to do both.
And, if the data is to be shown here, the M&M section must clearly state how this was done.
Authors' response to #3
"Data on this is still lacking. This study does not provide enough results to be able to draw sufficient conclusions. Nevertheless, we think the idea is worth a paragraph within this paper to encourage future investigations. We added “and to this point no sufficient conclusions can be drawn from this data”
No. If there isn't data to support conclusions, it is only possible to speculate. There must be a clear line between conclusions and speculations.
•
In summary, I tend to believe that this ms would require more work.